# Inverse designed plasmonic metasurface with parts per billion optical hydrogen detection

Ferry Anggoro Ardy Nugroho [1,2,5] ✉, Ping Bai[3,5], Iwan Darmadi [4,5], Gabriel W. Castellanos [3], Joachim Fritzsche [4], Christoph Langhammer [4] ✉, Jaime Gómez Rivas [3] ✉ & Andrea Baldi [1] ✉

Plasmonic sensors rely on optical resonances in metal nanoparticles and are typically limited by their broad spectral features. This constraint is particularly taxing for optical hydrogen sensors, in which hydrogen is absorbed inside optically-lossy Pd nanostructures and for which state-of-the-art detection limits are only at the low parts-per-million (*ppm*) range. Here, we overcome this limitation by inversely designing a plasmonic metasurface based on a periodic array of Pd nanoparticles. Guided by a particle swarm optimization algorithm, we numerically identify and experimentally demonstrate a sensor with an optimal balance between a narrow spectral linewidth and a large field enhancement inside the nanoparticles, enabling a measured hydrogen detection limit of 250 parts-per-billion (*ppb*). Our work significantly improves current plasmonic hydrogen sensor capabilities and, in a broader context, highlights the power of inverse design of plasmonic metasurfaces for ultra-sensitive optical (gas) detection.

Resonant optical sensors typically rely on wavelength shifts ($\Delta\lambda$) of their spectral features, such as peaks in transmission[1,2] and reflection[3,4], induced by analytes. To allow an accurate determination of the peak position and its quantitative dependence on the analyte concentration, one requires sensors with high-quality factors (*Q*-factors)[5,6], defined as the ratios of their resonance frequency by the corresponding linewidth. Since *Q*-factors are inversely related to the linewidth, which represents the losses of the resonant system, numerous strategies have been proposed to reduce these losses, and therefore sharpen the resonances and decrease the readout noise of optical sensors[5,7]. These approaches include the use of low-loss materials[8,9] and the tailoring of the resonator geometry as in nanoparticles-on-mirror[10], whispering-gallery-mode microcavities[11], and periodic metal nanoparticle arrays[12,13]. In particular, periodic nanoparticle arrays achieve high *Q*-factors by two processes: first, they reduce the radiative losses of individual nanoparticles by destructive interference of the coherently scattered radiation by the nanoparticles in the array; second, they redistribute the electromagnetic field into the surroundings, thus outside the individual metallic nanoparticles where losses originate[9,12,14–16]. This second condition benefits sensors probing phenomena occurring outside the metal nanoparticles, such as changes in the refractive index of the surrounding medium[17]. On the other hand, the removal of the field from the metallic nanoparticles is unfavorable for other classes of plasmonic sensors that probe changes inside the metal; the so-called direct plasmonic sensors.

Prominent examples of direct plasmonic sensors are plasmonic hydrogen sensors based on palladium (Pd) nanoparticles and their alloys[2,18–21]. These devices feature spark-free and room-temperature operation, efficient remote readout with small footprints, subsecond response time with excellent resistance to cross-contaminating and

[1]Department of Physics and Astronomy, Vrije Universiteit Amsterdam, De Boelelaan 1081, 1081 HV Amsterdam, The Netherlands. [2]Department of Physics, Universitas Indonesia, 16424 Depok, Indonesia. [3]Department of Applied Physics and Eindhoven Hendrik Casimir Institute, Eindhoven University of Technology, P.O. Box 513, 5600 MB Eindhoven, The Netherlands. [4]Department of Physics, Chalmers University of Technology, 412 96 Göteborg, Sweden. [5]These authors contributed equally: Ferry Anggoro Ardy Nugroho, Ping Bai, Iwan Darmadi. ✉e-mail: ferryanggoroardynugroho@yahoo.com; clangham@chalmers.se; j.gomez.rivas@tue.nl; a.baldi@vu.nl

deactivating gases, and long-term stability[2,22–24]. Mechanistically, these sensors rely on the barrierless dissociation of $H_2$ molecules at the surface of Pd nanoparticles and the subsequent intercalation of H atoms into the metal lattice. The corresponding change in dielectric function between pure Pd and Pd hydride leads to shifts in the localized surface plasmon resonance (LSPR) spectra of Pd nanoparticles, which are linearly proportional to the hydrogen concentration inside the particles[19,25]. Unfortunately, due to the lossy nature of palladium, the LSPR of Pd nanoparticles is broad[26,27], with full-widths at half maximum (FWHMs) typically >300 nm for nanostructures with plasmonic spectra at visible or near-infrared frequencies. Consequently, these broad peaks introduce inaccuracies in the determination of the sensing readout peak position, $\lambda_{peak}$[18], leading to higher signal noise, $\sigma$, and thus higher limits of detection (LoD), defined as the lowest analyte concentration measurable with a signal larger than $3\sigma$[7,28]. In fact, the detection limit still remains a significant challenge for plasmonic (and optical) sensors, with the state-of-the-art only at single-digit ppm; a comparably inferior performance than electrical sensors where ppb detection limit has been reported[29–33] (Supplementary Table 1). While ppm hydrogen sensitivity is appropriate for some applications, an ultralow detection limit, coupled with the abovementioned advantages of plasmonic sensing, is crucial for various application requiring local and early detection, such as hydrogen embrittlement in engineering structural materials[34], and intragastric hydrogen production in some cases of bacterial infections[35,36].

Here, we overcome the sensitivity bottleneck of optical hydrogen sensors by designing and experimentally demonstrating a sensor capable of detecting hydrogen gas down to the ppb level. Our sensing platform is based upon 2D periodic arrays of Pd nanoparticles that support collective surface lattice resonances (SLRs). These resonances emerge via the hybridization of the LSPRs of the individual nanoparticles and the constructive in-plane diffraction orders of the incoming light, known as Rayleigh anomalies (RAs)[12,37,38]. Since RAs emerge from interference effects outside the metal nanoparticles, they are characterized by narrow spectral features[39] that are therefore inherited by the SLRs. We employ inverse nanophotonic design—an algorithmic technique to find optical structures with set functional targets[40,41], to find sensor array configurations with the highest figure-of-merit (FoM), defined as the ratio between the SLR wavelength shift, $\Delta\lambda_{peak}$, and its FWHM. Critically, we find that the maximum FoM emerging from our evolutionary algorithm is not achieved by the array with the narrowest resonance, but rather by the array with an optimal balance between a narrow SLR and sufficiently large field enhancements inside the nanoparticles. This generic approach, which can benefit any direct plasmonic sensing platform, guides us to identify and experimentally demonstrate a sensor nanoarchitecture with a discernible signal down to 250 ppb; the lowest detection limit reported for an optical hydrogen sensor.

## Results

### Surface lattice resonances in Pd nanoparticle arrays

Despite their sensing potential, plasmonic SLRs have so far only been extensively studied on prototypical plasmonic metals, such as Au and Ag, with sensing applications limited to refractive index changes outside or at the surface of the metal[7,12,42–44]. The use of SLRs for direct plasmonic sensing of phenomena occurring inside the metals requires the utilization of active plasmonic metals such as Y[45], Mg[46], and Pd[27]. Hence, as a crucial step toward our optimized plasmonic sensor, we first demonstrate the existence of SLRs in periodically arranged Pd nanoparticles and characterize their optical spectra and field distributions. To this end, we fabricated an extensive set of square arrays of 45 nm high Pd nanodisks with varying diameters ($d$ = 70–180 nm, steps of 20/30 nm) and pitch distances ($a$ = 300–600 nm, steps of 50 nm) on fused silica ($n_{sub}$ = 1.46). To allow efficient radiative coupling between the nanodisks by the in-plane diffraction orders, an index-

matching medium is essential[38]. We thus coated the arrays with a 200 nm-thick poly(methylmethacrylate) (PMMA) film ($n_{PMMA}$ = 1.48). Besides having a suitable refractive index, PMMA is also serendipitously beneficial for Pd hydrogen sensors because it accelerates sorption kinetics by lowering the $H_2$ absorption activation energy into the Pd lattice and rebuffs other interfering and deactivating gases, such as $O_2$, CO, $NO_2$, and volatile hydrocarbons[2,47].

Figure 1 shows the experimental extinction spectra of 42 Pd nanodisk square arrays (corresponding scanning electron microscopy, SEM, images in Supplementary Fig. 1) alongside finite-difference time-domain (FDTD) calculations (see the "Methods" section) that accurately reproduce all spectral features in the measured data. In particular, we observe extinction spectra with one, two, or three peaks, depending on the nanodisk diameter and array pitch. The figure also includes the calculated extinction spectra of the corresponding single-particles, highlighting how arranging the nanodisks in periodic array results in distinct optical properties compared to their isolated counterparts. Particularly, as the pitch of the array, $a$, increases, the "main" peaks (as referenced to the array with $a$ = 300 nm) universally redshift and narrow (Supplementary Figs. 2 and 3). For example, for the $d$ = 180 nm samples, their FWHMs reduce by one order of magnitude from ~650 to ~65 nm. Furthermore, we observe the appearance of additional redshifting peak(s) at the lower wavelength(s) when $a$ reaches 350 and 500 nm. Last, scrutinizing closely these different peaks as a function of diameter reveals contrasting behaviors. While "main" peaks redshift as $d$ increases, the peaks at lower wavelengths are relatively immobile (Supplementary Fig. 2). This observation hints that the "main" peaks are SLRs that are dominated by contributions from the LSPR, and that the peaks at lower wavelengths are dominated by RAs, since their position depends solely on the particle-to-particle distance and not on the nanodisk diameter.

To further characterize the nature of the extinction peaks, we mapped the field distribution and optical dispersion relation of the array with $a$ = 550 nm and $d$ = 180 nm, which pronouncedly features three extinction peaks (Fig. 2a, see also a similar analysis for $a$ = 300 and 400 nm in Supplementary Figs. 4 and 5). Figure 2b depicts the FDTD-calculated field distribution map of the array at three different $y$-polarized excitation wavelengths corresponding to each of the peak wavelengths ($\lambda_1$–$\lambda_3$). At the two shorter wavelengths ($\lambda_1$ and $\lambda_2$), complex field distributions are found, with maxima located between the nanodisks. This feature suggests a strong contribution from the RA to the resonance, consistent with the discussion above. The excitation at the longest wavelength ($\lambda_3$) gives rise to field maxima in the vicinity of the nanodisks, suggesting a strong contribution of the LSPR to the resonance[48]. Given such field distribution, this relatively localized but narrow peak will emerge as the sensing peak for direct plasmonic sensing of hydrogen in our Pd nanodisk arrays.

After confirming the physical origin of the multiple peaks of our array, we carried out an angle dispersion extinction measurement (see the "Methods" section) and plot it alongside data from FDTD simulations (Fig. 2c). From the data we can determine the different RA orders that give rise to the optical properties of the array (see the calculation of Rayleigh anomalies in the Supplementary Information). In particular, the spectral shape is influenced by the higher (±1, ±1) modes at shorter wavelengths and by the (±1, 0) and (0, ±1) modes at longer wavelengths. The latter lower-order modes overlap with the LSPR deduced from the single-particle extinction peak at certain illumination angles. The coupling of the LSPR to the diffraction orders results in the narrowing of the resonance, while, as we show later below, maintaining its direct plasmonic sensing properties when exposed to $H_2$ gas.

### Inverse design of array parameters with highest figure-of-merit

Having established the ability to efficiently engineer the FWHM via SLRs in Pd arrays, we move on to design our hydrogen sensor with the

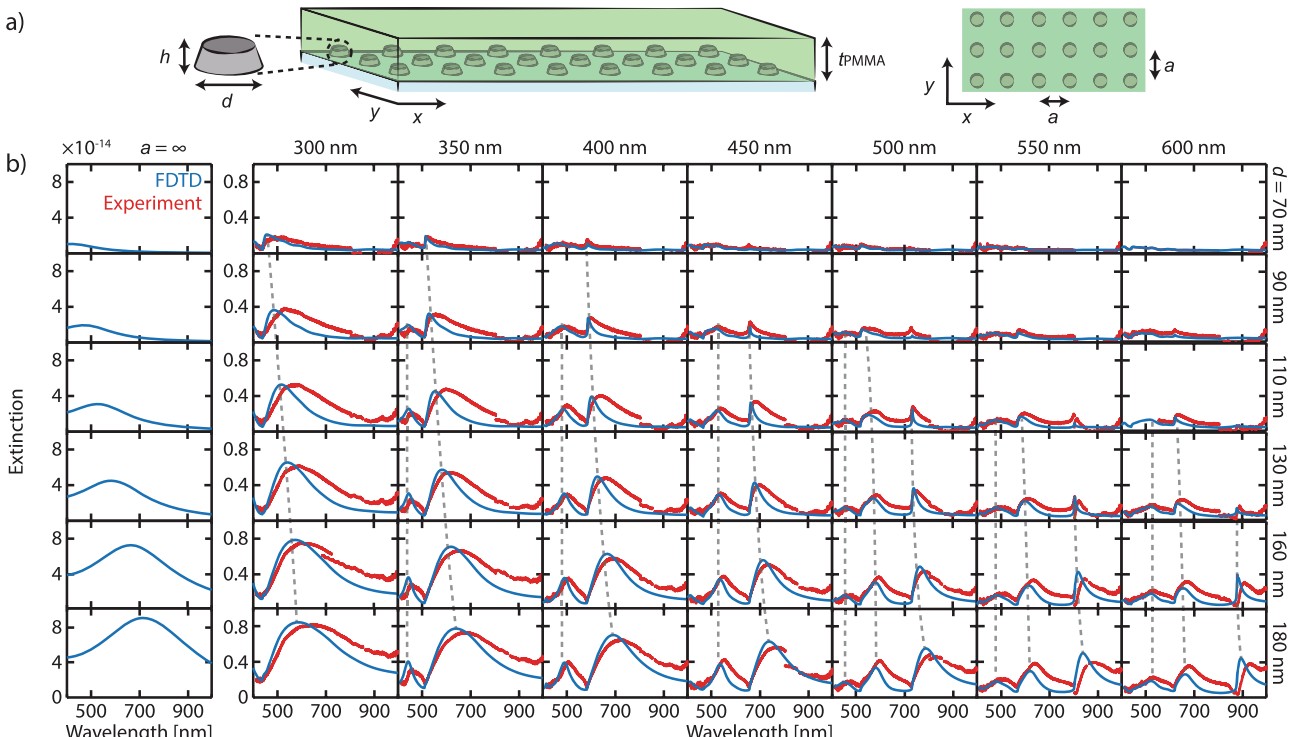

**Fig. 1 | Optical extinction spectra of Pd nanoparticles in a periodic array.**
**a** Schematic of the sample. **b** A collage of experimental (red) and simulated (blue) extinction spectra at normal-incidence from Pd nanodisk square arrays of fixed height, $h$, of 45 nm and PMMA layer thickness, $t_{PMMA}$, of 200 nm. The pitch of the array, $a$, increases from left to right from 300 to 600 nm. The nanodisk diameter increases from top to bottom from 70 to 180 nm. For comparison, the leftmost panels plot the extinction cross-sections (in m²) of the corresponding isolated single particles ($a = \infty$). Arranging the nanodisks in periodic arrays results in distinct optical spectra compared to their isolated single particle counterparts. The spectra comprise peaks originating from hybrid RA-LSPR modes—the SLR, see main text for details. Gray dashed lines are a guide to the eye to the position of corresponding extinction peaks as a function of particle diameter.

aid of FDTD calculation coupled with an inverse design optimization algorithm. As an optimization parameter for the performance of our sensor we use an FoM defined as (Fig. 3a)

$$\text{FoM} = \frac{\lambda_{\text{peakPdH}_x} - \lambda_{\text{peakPd}}}{\text{FWHM}_{\text{Pd}}}. \tag{1}$$

Our optimization aims at developing sensors capable of detecting $H_2$ gas at sub-ppm concentrations. At these concentrations and at room temperature, the Pd–H system will be in the so-called α-phase, characterized by hydrogen concentrations in the metal typically lower than ~1 at.%. To best model the optical properties of the palladium hydride phase, we, therefore, use the composition $PdH_{0.125}$, corresponding to the lowest $PdH_x$ composition for which an accurate dielectric function is available in the literature[49]. From Eq. (1), it is apparent that the highest FoM is obtained by finding an array configuration where the contributions from the LSPR (maximizing $\Delta\lambda_{\text{peak}}$ upon hydrogenation) and RA (narrowing FWHM) in their hybridized modes are optimized.

Typical plasmonic (hydrogen) sensing setups measure transmission or reflection spectra of the nanofabricated samples within the visible and near-infrared range (400–1000 nm). To have sensors with resonances within this range, we thus limit our array parameter searching space to $d = 100$–300 nm, $h = 20$–100 nm, and $a = 300$–500 nm (Fig. 3b). For the PMMA layer thickness, $t_{PMMA}$, we limit the range to 100–300 nm. The lower limit of 100 nm is chosen to ensure robust PMMA deposition when translating the parameters into a real sensor and also to provide a sufficient refractive index medium, while the upper limit of 300 nm is to avoid significantly slowing the $H_2$ diffusion kinetics introduced by a thicker layer[50].

To efficiently pinpoint the structural parameter combination with the highest FoM within such four-dimensional searching space, we adopt a particle swarm optimization[51] (PSO) algorithm combined with our FDTD calculations (Fig. 3b). This computational technique comprises populations that together assess the parameter space, and subsequently influence each other to move within this space to maximize the set goal (fitness parameter) that, in our case, is to maximize the FoM. We utilize 10 populations that start with a random set of parameter values and assess their corresponding FoMs. In the following generations, each population moves to other parameter values that result in a higher FoM (Fig. 3c). Running this process for 15 generations (see the "Methods" section), we move from an average FoM of 0.03 to 0.09, with a single-best population reaching 0.11. The corresponding optimized sensor architecture is $d = 124$ nm, $h = 20$ nm, $a = 376$ nm, and $t_{PMMA} = 300$ nm (Fig. 3d), with $\Delta\lambda_{\text{peak}}$ and FWHM of 32 and 296 nm, respectively (see other populations in Supplementary Table 2 and Supplementary Fig. 6). Looking at the optimized sensor extinction spectrum (Fig. 3d), it is interesting to note that only the LSPR-dominated peak responds to hydrogen, whereas the other peaks with lower LSPR contributions are less sensitive to changes in the refractive index of the nanodisks (Supplementary Fig. 7). This finding further corroborates our interpretation of the origin of the SLR peaks above. Finally, to appreciate the role of SLR excitation in obtaining sensors with high FoM, we also calculate the optical spectra of the optimized sensors single-particle counterparts (i.e., similar nanodisk parameters but not in array). As shown in Supplementary Fig. 8, the isolated nanodisk features comparable $\Delta\lambda_{\text{peak}}$, but suffers from an expansive FWHM of 498 nm, which drops its FoM to 0.07.

We also numerically assess the FoM for array parameters in close proximity to the ones of the optimized sensor architecture. In particular, we vary the pitch of the array, $a$, and the diameter of the

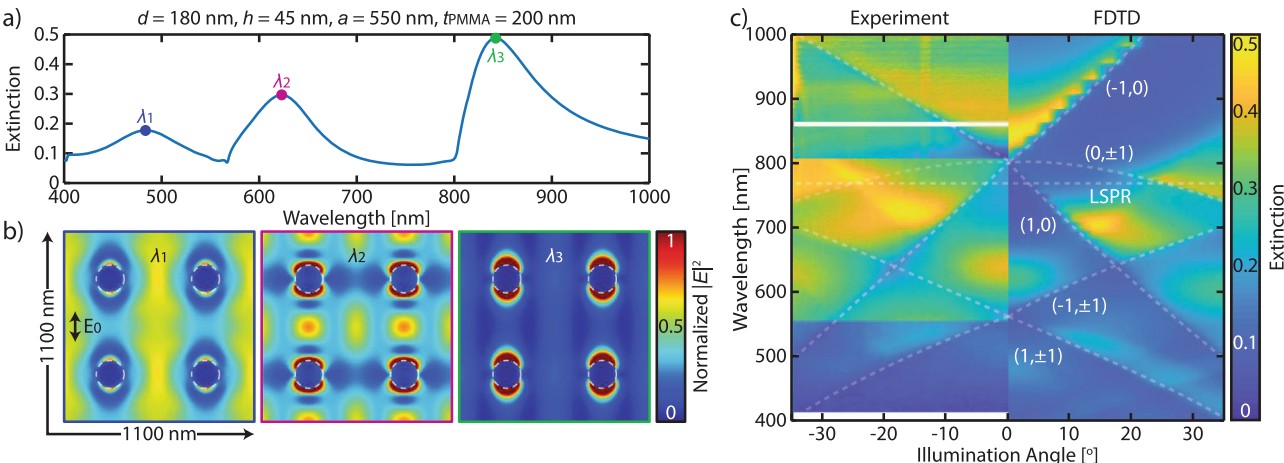

**Fig. 2 | Field distribution and optical dispersion relation of Pd nanoparticles arranged in a periodic array. a** Extinction spectrum of an array sample with $d = 180$ nm, $h = 45$ nm, $a = 550$ nm, and $t_{PMMA} = 200$ nm. **b** 2D maps of the normalized total field amplitude $|E|^2$ of the array at the mid-height of the nanoparticles and at different excitation wavelengths, as marked in panel (**a**). Excitation at wavelengths $\lambda_1$ and $\lambda_2$ generate field maxima far away from the nanodisks—a prominent characteristic in a RA mode. In contrast, excitation at $\lambda_3$ features field maxima

surrounding the nanodisks. Dashed lines outline the base of the nanodisks. **c** Experimental and simulated wavelength-resolved optical dispersion represented as the extinction spectra of the array for different angles of incidence. The dashed lines indicate the different RA orders of the array. The LSPR wavelength of the corresponding single-particle counterpart is also plotted which crosses two lower RA orders.

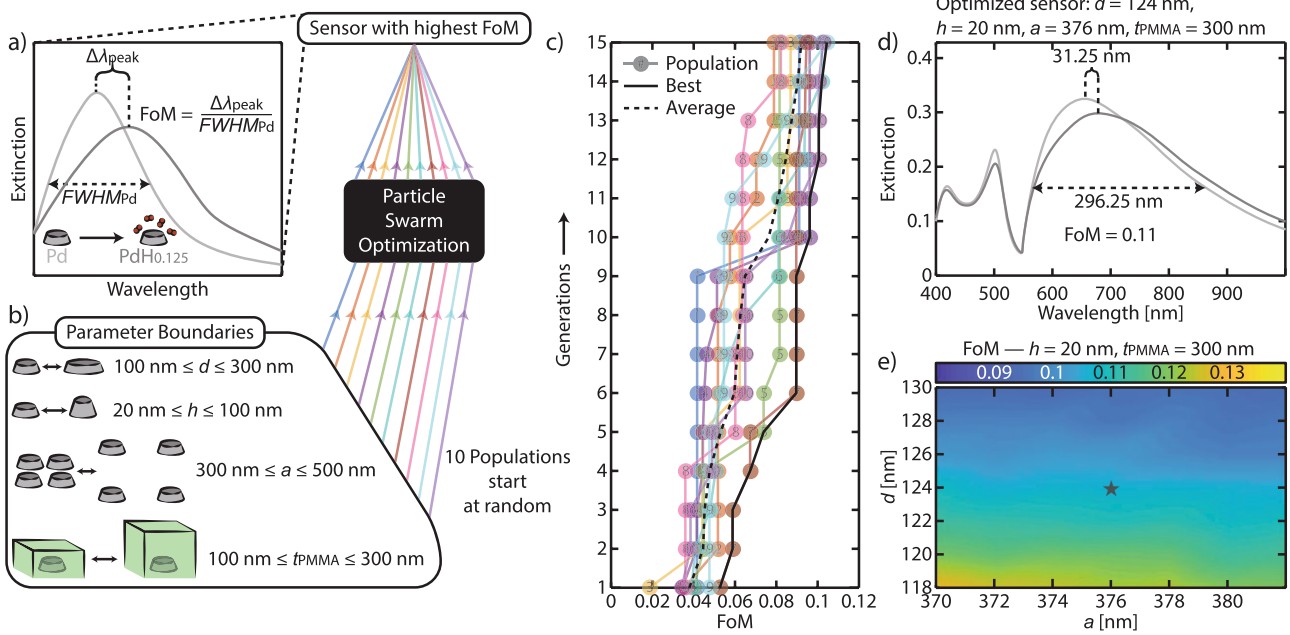

**Fig. 3 | Finding the sensor parameters with the highest FoM through particle swarm optimization. a** Schematic of the working principle and the associated figure-of-merit, FoM, of our plasmonic sensor. **b** Sketches of the four parameters defining the architecture of the Pd nanodisk arrays and their range used for the particle swarm optimization (PSO) algorithm. In this four-dimensional searching space, 10 populations are generated at random and let evolve iteratively through the PSO algorithm to find a sensor with the highest FoM, as defined in (**a**). **c** Evolution of the FoM for each of the 10 populations through 15 iterative generations. Clearly, in each generation, each population finds structures with higher

FoM. At the end, one of the populations reaches the highest FoM of 0.11. **d** Extinction spectra of the optimized sensor ($d = 124$ nm, $h = 20$ nm, $a = 376$ nm, $t_{PMMA} = 300$ nm) calculated for Pd (light gray) and PdH$_{0.125}$ (dark gray) nanodisk arrays. **e** Calculated FoM of nanodisk arrays with particle diameters $d$ and array pitches $a$ in close proximity to the ones determined for the optimized sensor (star symbol). The FoM exhibits ~10% variance from the optimized sensor, which indicates that a rather constant FoM can be achieved during the fabrication of the sensor.

nanodisks, $d$, within ±6 nm, as these are the parameters that are prone to the largest uncertainties in real sample fabrication via electron-beam nanolithography (see the "Methods" section). As shown in Fig. 3e, the FoM variation within the studied range is relatively small (0.09–0.13, ~10% from 0.11), which guarantees us to obtain the expected sensitivity when translating the optimized sensor parameters

into a real sample. Furthermore, it is clear that there are actually $a$ and $d$ combinations that result in slightly higher FoM, which could be identified if the PSO generation iteration would be expanded beyond 15 generations. However, there exists a complex relationship between small structural changes in our arrays and peak positions and line-widths in the corresponding extinction spectra. Given the relatively

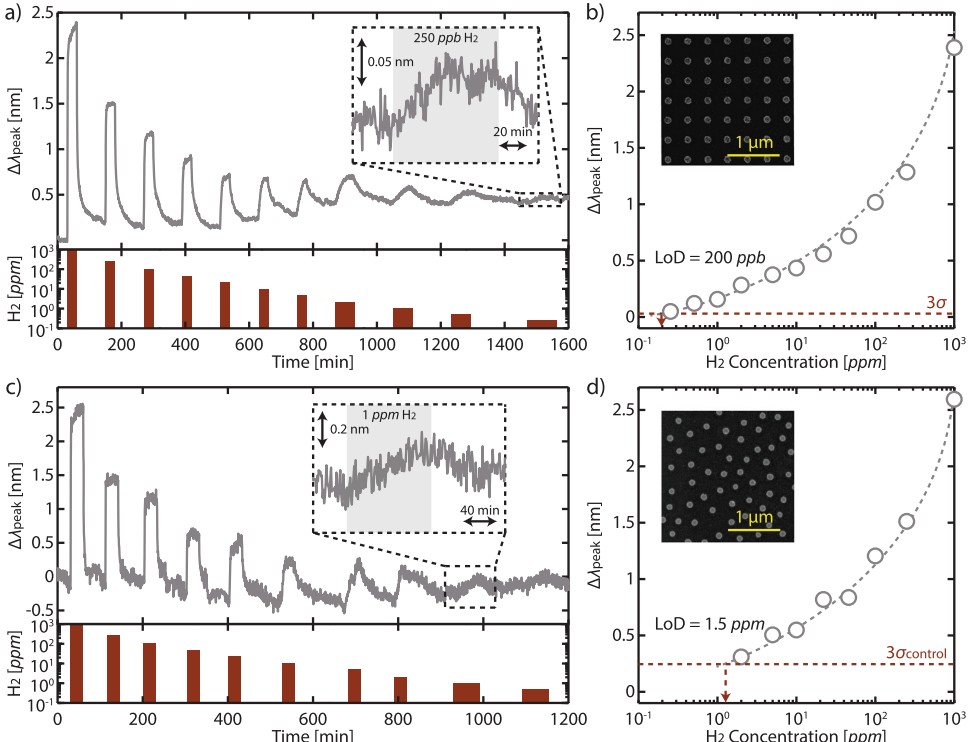

**Fig. 4 | Ultralow detection limit with Pd periodic array sensor. a** $\Delta\lambda_{peak}$ response to stepwise decreasing $H_2$ concentration (1000–0.25 ppm) in Ar carrier gas at room temperature. Inset: zoomed-in version of the sensor response to 250 ppb $H_2$. The slight baseline drift likely arise from minor adjustments in the setup over the course of the experiment[70]. **b** Measured $\Delta\lambda_{peak}$ as a function of $H_2$ concentration derived from (**a**). Gray dashed line is a guide to the eye and extrapolates the sensor response to the $3\sigma$ value (0.03 nm, red dashed line), indicating a LoD ~200 ppb (red dashed arrow). Inset: SEM image of the fabricated sensor. **c** and **d** are the data from a control quasi-random array sensor analogous to (**a**) and (**b**). The control sensor responds comparably to the periodic array sensor but suffers from its higher noise. Hence, its LoD is ~8 times higher at 1.5 ppm.

simple definition of our optimization parameter, FoM, extending our algorithm routine to more than ~15 generations typically led to coalescence of peaks and thus spuriously high FoMs originating from the inaccurate assignment of peak positions (Supplementary Fig. 10). While beyond the scope of the present work, these issues can be mitigated by a more rigorous definition of the FoM, peak, and the linewidth in the resulting optical spectra, by more stringent boundaries on the structural parameters of the nanodisks, and by using a dielectric function of much smaller hydride concentration relevant to the targeted $H_2$ concentration range of the sensor application.

### Realization of Pd nanoparticle array for hydrogen detection at ultralow concentration

Guided by the PSO results, we experimentally realized the optimized sensor design using electron beam lithography (Fig. 4a, b, Supplementary Fig. 11, and see the "Methods" section) and assessed its detection limit to $H_2$. To this end, we exposed the sensor to pulses of gradually decreasing $H_2$ concentration in Ar carrier gas (1000 ppm to 250 ppb, the lowest attainable concentration in our setup) at room temperature and plot its associated $\Delta\lambda_{peak}$, which is obtained through a Lorentzian fit[2] (Supplementary Fig. 12 and see the "Methods" section). As depicted in Fig. 4a, the sensor responds positively to different $H_2$ concentrations, with signal noise, $\sigma_{sensor}$, of 0.01 nm (Supplementary Fig. 12). Due to this small noise, the sensor is able to measure even the lowest 250 ppb pulse (Supplementary Fig. 14), making it the state-of-the-art optical hydrogen sensor to achieve sub-ppm detection (Supplementary Table 1). Recalling LoD as the lowest hydrogen pressure measurable with a signal larger than $3\sigma$, we extrapolate it to be ~200 ppb (Fig. 4b). Supplementary Figs. 15 and 16 show other performance aspects of the sensor, that is, selectivity and response/recovery

time, respectively. Regarding response/recovery times, the use of a thick PMMA film in our sensor significantly slows down the kinetics, with a response time of ~40 min at 250 ppb. However, as we discuss in the Supplementary Information and also later below, such speed can be increased by incorporating materials with faster hydrogenation kinetics and/or optimized nanostructure geometries.

As an important control, we fabricated an array with similar geometry parameters ($d = 124$ nm, $h = 20$ nm, and $t_{PMMA} = 300$ nm), but with the nanodisks dispersed quasi-randomly over the substrate rather than in a periodic lattice (Fig. 4c, d and Supplementary Fig. 18). We compared the optical response of this control sensor exposed to $H_2$ pulses under similar experimental conditions as the periodic array. Consistent with the FDTD simulations (Supplementary Fig. 8), the control sensor exhibits comparable $\Delta\lambda_{peak}$ with respect to the $H_2$ concentration (Fig. 4c). However, due to its larger FWHM, its $\lambda_{peak}$ determination results in a significantly higher noise, $\sigma_{control}$, of 0.08 nm (Supplementary Fig. 19), which ultimately leads to a LoD of 1.5 ppm, nearly an order of magnitude higher than the detection limit of its array sensor counterpart (Fig. 4d). This comparison accentuates the critical impact of the narrow FWHM, here engineered through the use of optimized SLRs, for resolving $\Delta\lambda_{peak}$ signals at low concentrations.

### Hydrogen detection limit in air

As a final step to demonstrate the applicability of our strategy in realistic gas environments, that is, to detect $H_2$ in air, we apply the concept of tandem polymers with different functionalities[2,52]. Such a concept allows us to utilize multiple (polymer) layers that independently provide targeted functionalities, such as to block $O_2$ molecules. For this purpose, we use poly(vinyl alcohol), PVOH, known for its very low $O_2$ permeability and thus widely used as an efficient $O_2$ barrier[53]. To

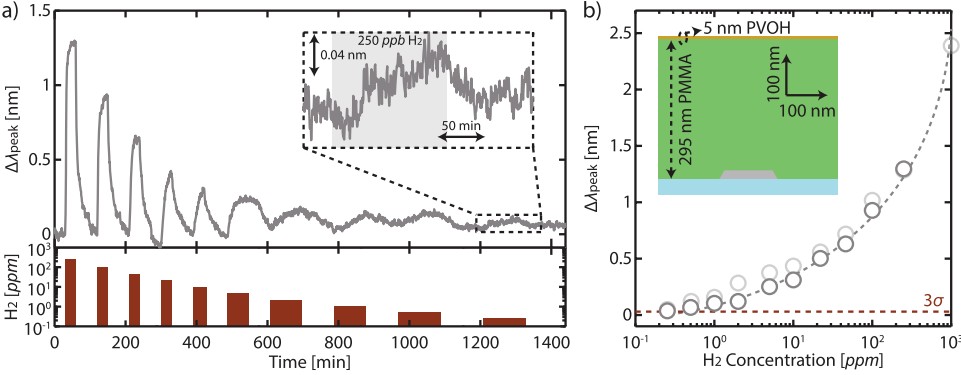

**Fig. 5 | Detection in air using a tandem polymer concept. a** $\Delta\lambda_{peak}$ response to stepwise decreasing $H_2$ concentration (1000–0.25 ppm) diluted in synthetic air at room temperature. Inset: zoomed-in version of the sensor response to 250 ppb $H_2$. **b** Measured $\Delta\lambda_{peak}$ as a function of $H_2$ concentration derived from (**a**). Gray dashed line is a guide to the eye. Light gray symbols are the response of the optimized sensor (i.e. without PVOH) in Ar (cf. Fig. 4b). The red dashed line marks the $3\sigma$ value (0.03 nm). Inset: To-scale schematic of the tandem sensor comprising a 5 nm PVOH film (orange), acting as an $O_2$ barrier, on a 295 nm PMMA layer (green). The tandem sensor shows a similar response to the optimized sensor measured in Ar.

maintain the optimized polymer thickness of 300 nm in our sensor, we etched the existing PMMA film by 5 nm (which was crucially required to deposit PVOH, see the "Methods" section) and subsequently compensated for such a loss with a 5 nm-thick PVOH layer (Fig. 5). Thanks to the similar refractive indices of PMMA and PVOH and the small PVOH layer thickness, the extinction spectra and the corresponding FoM of the tandem sensor are practically identical to the ones of the optimized sensor coated only by PMMA (Supplementary Fig. 17). Consequently, the tandem sensor exposed to decreasing $H_2$ concentration (1000 ppm to 250 ppb) diluted in synthetic air exhibits a very similar response to the optimized sensor in Ar (Fig. 5).

## Discussion

In summary, we have used an inverse nanophotonic design approach to identify and experimentally demonstrate an ultrasensitive plasmonic hydrogen sensor based on collective resonances in periodic arrays of palladium nanoparticles. The optimized sensor displays a non-trivial balance between a large optical response upon hydrogenation and narrow spectral features. The measured ppb limit of detection is an order of magnitude lower than any previous optical hydrogen sensor and becomes competitive with the more mature electrical sensors (Supplementary Table 1). The genericity of our strategy allows it to be combined with other optimization approaches, including the use of more sensitive transduction materials such as PdAu[2,19,25,54], (eightfold more sensitive than Pd at low $H_2$ concentrations) or PdTa[55] alloys, advanced data fittings capable of producing lower signal noise[56], and with sensor designs aimed at increasing detection speed such as the use of nanoparticles with faster $H_2$ sorption kinetics (e.g. PdAu[2,19,25,54], PdCo[24], PdTa[55]) and of coating layers with higher kinetic-enhancement effects (e.g. PTFE[2], twice as high as PMMA). Furthermore, we have so far only explored a simple and generic figure-of-merit parameter during our optimization, namely the peaks shift divided by the linewidth. Our inverse design approach, however, also permits the optimization of nanoparticle arrays for sensing platforms using different configurations such as perfect absorbers[20] and nanoparticles-on-mirror[57], and different readouts such as single-wavelength mode devices[58,59], opening the door to low-cost, practical[60], ultrasensitive platforms. Beyond hydrogen sensing, our approach can be extended to arrays of surface-functionalized nanoparticles with resonances that are sensitive to the adsorption of specific gasses via refractive index effects or chemical interface damping[61,62], with the potential to address a wider range of societal needs, from home safety to urban air pollution monitoring[63].

## Methods

### Sensor fabrication and characterization

The samples of Pd periodic array were fabricated from a 4-inch fused silica wafer with electron-beam lithography, thermal evaporation, electron-beam evaporation, wet-chemical etching, reactive-ion etching, lift-off, and dicing. The steps involved included: (i) using a 4-inch fused silica substrate (Siegert Wafer), a lift-off layer of 80 nm MCC NANO Copolymer EL4 (Microlithography Chemicals Corp.) was first spin-coated and baked on a contact hotplate for 5 min at 180 °C. Following that, an imaging layer of 70 nm MCC NANO 950k PMMA A2 (Microlithography Chemicals Corp.) was spin-coated and baked on a contact hotplate for 5 min at 180 °C. (ii) A 20 nm-thick Cr layer was deposited with thermal evaporation (Lesker Nano 36) to enable electrical discharge during electron-beam exposure. (iii) The nanodisks were defined in the double resist layer on areas of $10 \times 10$ mm$^2$ with electron-beam lithography (Raith EBPG 5200) by exposing circles of 35 nm radius. Each circle was filled with 19 shots at a beam current of 50 nA and at a base frequency of 5.19 MHz. (iv) The 20 nm Cr discharge layer was removed by immersing the substrate for 60 s in Nickel/Chromium etchant (SunChem), followed by water rinsing and blow drying. (v) The exposed resist was developed for 60 s in MIBK 1:3 IPA solution, dried in $N_2$-stream, and descummed in oxygen plasma for 5 s at 50 W RF-power, 250 mTorr chamber pressure, and 40 sccm gas flow in a BatchTop Reactive Ion Etcher (PlasmTherm). (vi) To form the nanostructures, Pd was deposited through the resist mask with electron-beam evaporation at a deposition rate of 1 Å s$^{-1}$ in a PVD 225 system (Lesker), lifted off in acetone for 24 h, rinsed in IPA and blow dried in $N_2$-stream. (vii) Finally, the wafer was diced (DAD3350, Disco) into individual chips of $10 \times 10$ mm$^2$. For the control quasi-random array sample, the fabrication procedures (steps, materials, and tools used) followed exactly the protocol reported in ref. 2, which is reproduced here for completeness (i) Fused silica substrates were cleaned by ultrasonic agitation consecutively in acetone, isopropanol, and de-ionized water. (ii) 950k A4 PMMA (Microlithography Chemicals Corp.) was spin-coated onto the substrates at 2000 rpm for 30 s (yielding a PMMA thickness of ~280 nm) followed by soft baking at 170 °C on a hotplate for 5 min. (iii) Samples were subjected to a 5 s oxygen plasma (50 W RF-power, 250 mTorr chamber pressure, and 40 sccm gas flow in a BatchTop Reactive Ion Etcher, PlasmTherm) to enhance the hydrophilicity of the sample surface. (iv) A polyelectrolyte solution (poly diallyldimethylammonium, Mw = 200,000–350,000, Sigma Aldrich, 0.2 wt% in Milli-Q water, Millipore) was pipetted on the surface of the samples and left to incubate for 40 s before rinsing in de-ionized water, creating a positively charged surface layer on the PMMA

surface. (v) A suspension of negatively charged polystyrene beads (PS, 120 nm sulfate latex, Interfacial Dynamics Corporation, 0.2 wt% in Milli-Q water) was added to the surface. The size of the PS beads determined the diameter of the fabricated nanodisks at the end of the processing. After 3 min incubation, the suspension was rinsed away with de-ionized water, and the samples were blown dry with nitrogen gas. (vi) A 15-nm-thick Cr film was evaporated using a Lesker PVD 225 Evaporator at a base pressure of $5 \times 10^{-7}$ torr and evaporation rate of $1 \text{Å s}^{-1}$. (vii) The PS beads were removed by tape stripping (SWT-10, Nitto Scandinavia). This left a Cr film with holes at the positions of the stripped PS beads. (viii) The samples were subjected to 5 min oxygen plasma treatment (50 W RF-power, 250 mTorr chamber pressure, and 40 sccm gas flow in a BatchTop Reactive Ion Etcher, PlasmTherm) to etch through the PMMA layer exposed beneath the holes in the Cr mask. (ix) Through this mask, Pd was deposited at a deposition rate of $1 \text{Å s}^{-1}$. (x) The remaining PMMA layer was dissolved in acetone in a liftoff step, removing the mask from the sample and leaving only the nanodisk structures on the substrate. (xi) Finally, samples were soaked in isopropanol and blown dry with nitrogen, resulting in quasi-random Pd arrays on fused silica substrate. Specific to this work, to deposit PMMA on top of the samples, a spin coat of 950k PMMA A4 (Microlithography Chemicals Corp.) was conducted, followed by soft baking on a hotplate for 5 min at 170 °C. To produce the tandem sample, the sensor with PMMA was first etched in oxygen plasma for 2 s (50 W RF-power, 250 mTorr chamber pressure, and 40 sccm gas flow in a BatchTop Reactive Ion Etcher, PlasmTherm), to introduce hydrophilicity to the surface so that the PVOH solution can be drop casted on it, which also resulted in the reduction of the PMMA thickness by -5 nm according to the etch rate determined before[64]. Following that, a PVOH solution (0.1 wt.% in water) was spin-coated (5000 rpm, 60 s) and then baked at 80 °C for 5 min. The obtained thicknesses were confirmed by ellipsometry (J.A. Woollam M2000). The SEM images were collected from glass samples coated with 5 nm Cr layer (Zeiss Supra 60 VP with secondary electron detector, working distance 4 mm, and an electron beam acceleration voltage of 7–15 kV).

## Finite-difference time-domain simulations

We used commercial Lumerical FDTD software to calculate the optical properties of both single and array Pd nanodisks. We modeled the nanodisks as cylinders with a taper angle (the angle between the base and the side wall) of 65° to be close to the fabricated samples[65]. The permittivity values of Pd and Pd hydride ($PdH_{0.125}$) were taken from the literature[49]. The nanodisks were placed directly on top of a fused silica substrate ($n = 1.46$). On top of the substrate and embedding the particles, a PMMA layer was added, whose permittivity was obtained from an ellipsometry measurement[64]. For the tandem sample, the permittivity of PVOH was also obtained from an ellipsometry measurement (J.A. Woollam M2000, Supplementary Fig. 17). Finally, on top of this layer, there was air ($n = 1$). The simulations of the scattering efficiencies were done using a total-field scattered-field (TFSF) source with a broadband (400–1100 nm) beam incident from the air and along the normal direction. The TFSF source divides the simulation region into two concentric volumes: one centered around the particle with the total fields, and another external where only the scattered fields propagate. Power transmission monitors were positioned around the TFSF source to calculate the scattering cross sections. The efficiency was calculated by dividing the former quantities by the geometrical cross-section, i.e., the area of the cylinder perpendicular to the propagation vector $k$ of the incident field. Perfectly matched layer (PML) boundaries were implemented in every direction. The simulations of the periodic arrays were performed using periodic boundary conditions in the $x$- and $y$-directions, and PML boundaries in the $z$-direction. The illumination consisted of a broadband (400–1100 nm) beam, approximated by a plane wave, which was incident normal to the array plane (the $xy$-plane) from the air. To extract the transmission, a $xy$

monitor was placed on the substrate side. Another $xy$ monitor was placed at the center of the particles to extract the fields. The polarization of the incident electric field was set along the $y$-axis. To extract the extinction (i.e. 1−transmission) dispersion data, several simulations with different incident angles were performed.

## Optical dispersion measurements

An unpolarized broadband light source was used to illuminate the samples and investigate their optical dispersion. The light was focused onto the sample and collected with a Nikon L Plan ×20/0.45NA and a Nikon S Plan Fluor ×40/0.6NA objectives, respectively. Using a dedicated lens system, the back focal plane of the objective was imaged with an imaging spectrometer connected to a multiplying CCD camera (ProEM: 512B). The back focal plane contained the Fourier transform of the optical field transmitted by the sample upon illumination, i.e. the angular dispersion of the transmitted light. The image on the CCD contained 2D angular information for all the wavelengths illuminating the sample. Closing the slit that controlled the light entering the imaging spectrometer allowed the selection of one angular component and its spectral decomposition into the CCD. To get accurate wavelength resolution, a grating of 150 g mm⁻¹ was used. This allowed ±150 nm range to be imaged for a selected wavelength center. To image the full spectrum of the sample (400–1000 nm) we measured spectra at several wavelength centers (i.e. 470, 620, 770, and 900 nm). Using a polarizer before the illumination objective allowed us to select between TM and TE polarizations.

## Particle swarm optimization calculation

To design the most sensitive hydrogen sensors, the FDTD method associated with the particle swarm optimization (PSO) algorithm was adopted. PSO is a robust population-based stochastic evolutionary computation technique, which is inspired by the natural social behavior and dynamic movements with communications of animal species (called particles) and looking for their requirements in a search space[51].

Here PSO was employed to optimize the structural parameters of the plasmonic hydrogen sensor to yield the highest FoM defined by Eq. (1). To this end, we chose to use $PdH_{0.125}$ for the calculation of the hydride phase for the following reasons: (i) This is the lowest Pd hydride concentration whose dielectric function is available in the literature[49]. (ii) At this concentration, the Pd hydride is still at the diluted $\alpha$-phase, with negligible lattice expansion. This condition prevents inaccurate calculation during FDTD simulation where the expansion of the nanodisk has to be included[66]. (iii) The chosen hydride concentration is also in line with the targeted range of the hydrogen concentration. (iv) Lastly, the accompanied spectral change of the sensor at this hydride concentration was expected to be small enough so that it would be the same SLR peak that was considered, thus avoiding false $\Delta\lambda_{peak}$ determination when calculating the FoM, as we detailed later below.

To begin the optimization, the algorithm was initialized with 10 Pd/$PdH_{0.125}$ nanodisk arrays of random locations of parameters in their own spaces, which then were sent to the Lumerical FDTD platform, where the transmission was numerically evaluated. After that, FDTD sent the computed optical values back to the algorithm where the FoM was calculated and produced the parameters for the next generation. The full technical description of the PSO used here is provided in the Supplementary Information.

Ideally, PSO should keep iterating until all particles converge to the global optimal solution instead of stopping at the 15th generation as in our present case here. However, we found that a number of populations updated their FoM through very large peak shifts and very broad FWHM that included two SLR different peaks. In Supplementary Fig. 10, we show an example of a population of the 18th generation. In this case, there are two close SLR peaks in both Pd and $PdH_{0.125}$ arrays at around 530 and 650 nm, respectively. In this case, the algorithm

wrongly considered the lower wavelength peak for the case of Pd, and the longer wavelength peak for the case of $PdH_{0.125}$, causing a wrong evaluation of $\Delta\lambda_{peak}$. Furthermore, the FWHM was also calculated for two close SLRs. Consequently, in such a case, we cannot correctly calculate the FoM since the parameters originated from two different peaks. Because this similar array began to appear in the 16th generation, we stopped the PSO after the 15th generation and confirmed that all the FOM calculated were from a single SLR peak (Supplementary Fig. 6).

## Hydrogen sensing measurements

Prior to measurements, all sensors were exposed to multiple cycles (>20) of pure $H_2$ (1 bar) and vacuum at room temperature to stabilize the hydrogen-induced microstructural changes in the nanoparticles[67]. The sensors' LoD determination was performed in a custom-made reactor chamber (effective volume ca. 1.5 mL) equipped with two fused silica viewports (1.33" CF Flange, Accu-Glass) that enabled transmission-mode optical monitoring. The detail of the chamber is reported in ref. 68. The transmission measurements were carried out through a fiber-coupled, unpolarized halogen light source (AvaLight-HAL-S-Mini) and a high-resolution visible range spectrophotometer (Avantes Sensline Avaspec-HS-TEC). The $H_2$ gas concentration was controlled by adjusting the flow rate ($\nu$ [mL min$^{-1}$]) ratio of 1000 ppm $H_2$ (diluted in Ar), 100% Ar and 100% synthetic air using mass flow controllers (MFCs, Bronkhorst El-Flow Select series, see Supplementary Table S3). All experiments were carried out at a constant 30 °C, regulated via a PID controller (Eurotherm 3216) in a feedback loop manner, where the sample surface temperature inside the chamber was continuously used as input. For the selectivity tests, the measurements were carried out in a quartz tube flow reactor with optical access for transmittance measurements[69] (X1, Insplorion AB). The gas flow rate of 350 ml min$^{-1}$ and gas composition were regulated by mass flow controllers (Bronkhorst $\Delta P$). The sample was illuminated by unpolarized white light (AvaLight-Hal, Avantes) with a coupled optical fiber with the collimating lens. The transmitted light was recorded using a spectrometer (AvaSpec-1024, Avantes). The measurement temperature was maintained at 30 °C and the chamber was kept at atmospheric pressure. As readout, the LSPR peak descriptors ($\lambda_{peak}$) were obtained following the method we established earlier[2]. In detail, a Lorentzian fit was applied to the wavelength range at ±60 nm around the LSPR peak in the measured optical extinction spectra. Despite the asymmetry of the global LSPR peak, a good fit ($R^2 > 0.97$) was obtained, and thus the fit is appropriate to determine the $\lambda_{peak}$ (Supplementary Figs. 12 and 19).

## Data availability

All experimental data within the article and its Supplementary Information are available from the corresponding authors upon request.

## Code availability

All code used within the article is available from the corresponding authors upon request.

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

## Acknowledgements

We acknowledge financial support from The Netherlands Organization for Scientific Research through the NWO Vidi Award 680-47-550 and Vici Award 680-47-628, the Swedish Foundation for Strategic Research Framework project RMA15-0052, the Knut and Alice Wallenberg Foundation project 2016.0210 and the Swedish Energy Agency Project

49103-1. F.A.A.N. acknowledges support from the European Union's Horizon 2020 research and innovation program under the Marie Skłodowska-Curie Grant Agreement No. 101028262. Part of this work was carried out at the Chalmers Micro- and Nanofabrication Laboratory MC2 and the Materials Analysis Laboratory (CMAL) under the umbrella of the Chalmers Excellence Initiative Nanoscience and Nanotechnology. We also thank Dr. Sven Askes and Dr. Ruben Hamans for the critical reading of the manuscript.

## Author contributions

F.A.A.N. and A.B. conceived the project. F.A.A.N. and P.B. executed the FDTD calculation. P.B. and G.W.C. developed and performed the PSO algorithm. I.D. and J.F. fabricated the sensors. F.A.A.N. and I.D. designed and performed the sensing measurements. F.A.A.N. and G.W.C. measured the transmission and angle dispersion of the arrays. C.L. supervised I.D. and J.F., J.G.S. supervised P.B. and G.W.C., A.B. supervised F.A.A.N. and the project as a whole. F.A.A.N. performed data analysis and wrote the first draft of the manuscript. All authors commented on and edited the manuscript.

## Competing interests

The authors declare the following competing interests: C.L. is co-founder of a company that markets nanoplasmonic sensor-based technologies.
