## [Peer Review File · Nature Communications]

Inverse Designed Plasmonic Metasurface with Parts per Billion Optical Hydrogen DetectionREVIEWER COMMENTS

Reviewer #1 (Remarks to the Author):

The work by Nugroho et al demonstrates a novel approach to overcome the relatively low sensitive of optical hydrogen sensors based on LSPR. Although Pd based nanoparticles for optical sensing of hydrogen (including plasmonic based) has been well studied, including by the authors, and inverse designed plasmonic metasurface is not novel. However, the unique application of this technique to realize a hydrogen detector with detection limits in ppb is quite novel. As someone who is well experienced in this field, I find this work very interesting, the paper is very well written and the results very relevant to broad community of scientists in the sensor committee and related fields. Therefore, I support the publications of this work in Nature communications. Some minor points to be addresses by the authors are:

1. It will be helpful to the reader if the architecture (diagram/layout) of the sensor is included.
2. It is not very clear (at least to me) from the experimental details how the 100-300 nm PMMA was deposited on the Pd nanoparticles.
3. Some important properties of a sensor are the response and recovery times, selectivity/cross sensitivity to other gases, reproducibility, and error in the readout (precision). However, the authors did not discuss these aspects satisfactorily. It will be good to show a clearer response and recovery times as a function of concentration. I do see about 20 min for 250 ppb (too high for a sensor), also the change in the $\Delta\lambda_{\text{peak}}$ as a function if H₂ concentration seems very small at low H₂ concentration to ensure a precise determination of the hydrogen concentration. How reproducible are the measurements (readout) upon several cycles of hydrogenation dehydrogenation? These properties should at least be addressed for publication in such a high impact journal.

Reviewer #2 (Remarks to the Author):

In this Communication, authors Nugroho et al. use an iterative algorithmic process to optimize an array-based hydrogen sensor. Tweaking parameters such as nanodisk diameter, nanodisk height, nanodisk pitch distance, and filtering layer (PMMA) thickness, the authors sought to optimize a figure of merit, and, by extension, minimize the hydrogen limit of detection (LoD). While the method of analysis is novel, and may be a significant contribution, this can not be concluded from the data presented here, as detailed below.

This work is not at the level of quality normally associated with Nature Commun. It may be publishable in a more specialized journal after major revision, including the addition of key data. The major issues are the following:

1). While low LoDs are potentially important in terms of understanding mechanistic aspects of sensor function, in any practical application required for hydrogen sensing, detection of the gas in a reactive atmosphere - such as air or humid air (in the case of safety sensors) - is absolutely required. In this paper, sensing was accomplished entirely with dry argon as the backing gas. Prior work has already demonstrated that air is a far more challenging medium for H₂ sensing than inert gases such as N₂ and Ar - especially as it concerns sensitivity and LoD. Papers, such as this one, that choose an expedient medium for sensor testing are routinely rejected from ACS Sensors, based upon the requirement that *meaningful media* for all types of sensors are an absolute requirement. This is simply not the case here. The authors write, "We expect such sensitivity to also hold in air, thanks to the excellent O₂ sieving provided by PMMA". These data should be added in revision.

2) Likewise, extrapolation from higher H₂ partial pressures to very low LoD values, which are then quoted as an LoD, is unacceptable. If the point of the paper is ppb sensitivity, then please demonstrate measurements in this concentration range with some degree of robustness (that is to say, multiple exposures of the sensor to these concentrations so that the reproducibility of the

response can be assessed). A 1 ppb LoD is mentioned at some point, and this is miles away from the value supported by the data provided in the ms.

3). A important issue with the presentation of the sensing data is that the exposures to H₂ are carried out *in order of decreasing concentration*. This practice obscures any "memory effect" of the sensing elements. It is well understood that proper practice is to carry out these exposures in random order of H₂ concentration, with the exposure to several concentrations across the calibration curve repeated more than once so that the reproducibility of the sensor response can be assessed - after exposures to a range of different H₂ concentrations. This practice of presenting a train of monotonically decreasing (or increasing) H₂ exposures (increasing common!) should be rejected by journals of the stature of the Nature journals.

4). Baseline drift is a major issue in the data presented in Fig. 4. Especially in dry Ar gas, this is surprising. What is the origin of this drift? Is it caused by H₂ exposures, or is the baseline drifting before H₂ exposures are initiated? Presumably, these data are amongst the best acquired by the authors, meaning that data sets NOT showing baseline drift were not available. The practical utility of any gas sensor exhibiting this characteristic is very limited.

Reviewer #3 (Remarks to the Author):

In this paper the authors used an inverse nanophotonic design approach to identify and experimentally demonstrate an ultrasensitive plasmonic hydrogen detector based on collective resonances in periodic arrays of palladium nanoparticles. The measured ppb limit of detection seems to be an order of magnitude lower than previously optical hydrogen sensor.

the idea to optimize the array configuration in order to achieve very high sensitivity is not new and has been reported recently in ACS Sens. 2020, 5, 4, 917–927 (not cited in this paper). The original approach used by the authors here comprised a population-based stochastic evolutionary computation technique. This sounds actually interesting because it can be potentially applied to many other applications.

The other noteworthy aspect/result in the paper is the obtained detection limit (sub ppm). Anyway, although this is an order of magnitude better with respect to other examples in literature, the way it is calculated is not convincing. The authors should report (or eventually perform) H₂ exposure also starting from the lower concentrations (i.e. from 0 up to 1000ppm). the delta lambda of 0.05nm (0.5 Ang!!) sounds very hard to be reproducible. moreover, it is known that H₂ exposure can change the volume of Pd nanostructures and this can impact in the resonance of the system (for this reason is important to test the system starting from 0 and increasing the concentration) (this effect is for example discussed in Optics Express 28, , pp. 25383-25391 (2020); and Journal of Alloys and Compounds 704, 2017, 303-310)

Finally, the authors should discuss how this value can be measured in a real on-field application.

With respect to the state-of-the-art, the authors missed to mention several important papers in the references (among the others - actually there are tons of papers on H₂ sensing by means of Pd nanostructures):

ACS Sens. 2020, 5, 4, 917–927
Nano Energy 71, 22020, 104558
Nanophotonics, vol. 3, no. 3, 2014, pp. 157-180
MRS Bulletin volume 38, 495–503 (2013)
Nature Materials volume 10, 631–636 (2011)

Sensors and Actuators B: Chemical 295, 15, 2019, 101-109
Nanomaterials 2021, 11, 3100

Below, we have addressed all their concerns and comments in detail in our point-to-point response. Our reply also provides the additional data requested by the reviewers. We are convinced that our response will serve as the basis for reaching a firm positive decision.

Point-to-Point Response

In this Letter, we reproduced the Reviewers' comments in **red**, and wrote our corresponding responses in **black**, and the referred/change in the text (both main text and SI) in **blue**.

Reviewer 1

The work by Nugroho et al demonstrates a novel approach to overcome the relatively low sensitive of optical hydrogen sensors based on LSPR. Although Pd based nanoparticles for optical sensing of hydrogen (including plasmonic based) has been well studied, including by the authors, and inverse designed plasmonic metasurface is not novel. However, the unique application of this technique to realize a hydrogen detector with detection limits in ppb is quite novel. As someone who is well experienced in this field, I find this work very interesting, the paper is very well written and the results very relevant to broad community of scientists in the sensor committee and related fields. Therefore, I support the publications of this work in Nature communications. Some minor points to be addresses by the authors are:

We thank the Reviewer for his/her very positive assessment regarding the novelty and presentation of our work, its potential broad implications, and recommendation for publication.

1. It will be helpful to the reader if the architecture (diagram/layout) of the sensor is included.

We now have included such layout as an inset in Figure 1, where the periodic array is first introduced.

2. It is not very clear (at least to me) from the experimental details how the 100-300 nm PMMA was deposited on the Pd nanoparticles.

Indeed, we missed to include this part in our manuscript. For such purpose, we use the standard spin coating process of PMMA. We have now added this portion in the relevant Methods section:

“To deposit PMMA on top of the samples, a spin coat of 950k PMMA A4 (Microlithography Chemicals Corp.) was conducted, followed by soft baking on a hotplate for 5 min at 170 °C. The obtained thicknesses were confirmed by ellipsometry (J.A. Woollam M2000).”

3. Some important properties of a sensor are the response and recovery times, selectivity/cross sensitivity to other gases, reproducibility, and error in the readout (precision). However, the authors did not discuss these aspects satisfactorily. It will be good to show a clearer response and recovery times as a function of concentration. I do see about 20 min for 250 ppb (too high for a sensor), also the change in the $\Delta\lambda_{\text{peak}}$ as a function of H₂ concentration seems very small at low H₂ concentration to ensure a precise determination of the hydrogen concentration. How reproducible are the measurements (readout) upon several cycles of hydrogenation dehydrogenation? These properties should at least be addressed for publication in such a high impact journal.

We thank the Reviewer for the relevant comment regarding other aspects of the sensors. Since in this manuscript we focused on the sensitivity, we initially excluded measurements related to speed and selectivity. Nonetheless, we agree that it will add value to the manuscript and hence we now added this data.

Related to selectivity, we now add experiments where we expose the sensors to H₂ mixed with CO and NO₂, the gases that typically poison the surface of Pd and render it inactive. From the Figure below, which we now add to the SI, it is clear that the PMMA efficiently filters these poisoning gases and thus the sensor maintains its response within 20% with respect to pure H₂ (panel b). This filtering effect of PMMA is not surprising since it has been reported before (see for example <https://www.nature.com/articles/s41563-019-0325-4> and <https://www.nature.com/articles/s41467-021-22697-w>). However, as we wrote in the manuscript, it is clearly an advantage of our sensor here as the PMMA (or other polymer coating) is an integral part of our design to excite the SLR and thus to obtain the high sensitivity.

Supplementary Figure 16. (a) Time-resolved $\Delta\lambda_{peak}$ response of sensor 1 pulse of 1000 ppm H₂ followed by 5 pulses of 1000 ppm H₂ + 500 ppm CO, and 1000 ppm H₂ + 50 ppm NO₂ in Ar. (b) Normalized sensor signal to the one obtained in 1000 ppm H₂. The error bars denote the standard deviation from 5 cycles. The shaded area indicates the $\pm 20\%$ deviation limit from the normalized $\Delta\lambda_{peak}$ in 1000 ppm H₂.

We refer this Figure in the main text as

“..., see also Supplementary Figures 15 and 16 for additional data on the sensor’s selectivity, response time, and recovery time.”

and since these selectivity measurements were done in a different flow reactor, we now have added the experiment details in the Methods.

“For the selectivity tests, the measurements were carried out in a quartz tube flow reactor with optical access for transmittance measurements (X1, Insplorion AB). Gas flow rate of 350 ml min⁻¹ and gas composition were regulated by mass flow controllers (Bronkhorst ΔP). The sample was illuminated by white light (AvaLight-Hal, Avantes) with a coupled optical fibre with collimating lens. The transmitted light was recorded using a spectrometer (AvaSpec-1024, Avantes). The measurement temperature was maintained at 30 °C and the chamber was kept at atmospheric pressure.”

With regard to sensor's speed, we now have analyzed quantitatively the corresponding response and recovery times of our sensors, as shown in the Figure below (now included in SI), which we refer in the main text as

“..., see also Supporting Figures 15 and 16 for additional data on the sensor's selectivity, response time, and recovery time).”

Supplementary Figure 15. The definition of (a) response time as t_{90} and (b) recovery times as t_{10} , which correspond to the time it takes to reach 90% and 10% of the normalized signal (with respect to signal during the exposure and in the absence of H_2), respectively. (c) Response times and (d) recovery times of the optimized periodic array sensor and control random array sensor as function of H_2 concentration. Data is extracted from Fig. 4a and c, respectively. The recovery and response times of both sensors are comparable and can practically be described with a single trend (the dashed lines), as established in ref. ¹⁷.

17. Nugroho, F. A. A. *et al.* Metal–polymer hybrid nanomaterials for plasmonic ultrafast hydrogen detection. *Nature Materials* **18**, 489–495 (2019).

We note that both response and recovery times (define as t_{90} and t_{10} , respectively, as defined in panel a and b) increase with lowering H_2 pressures. This effect is inherent for Pd-H system and has been observed consistently in other works (for example in <https://www.nature.com/articles/s41563-019-0325-4> and <https://www.nature.com/articles/s41467-021-22697-w>).

Consequently, for our lowest H_2 concentration of 250 *ppb*, the Reviewer is correct that the response time is around 40 min. We acknowledge that this is very slow and not an ideal case for real application. Nonetheless, the key message in this manuscript is that

our method of increasing the sensitivity is independent of the direct sensing platform used. In fact, when comparing the optimized periodic sensor with its corresponding control random array (Figure 4c in the manuscript), both sensors have *comparable* speed and can practically be described with a single trend. In other words, our method of increasing sensitivity through arranging the sensor in a periodic array *does not affect* the speed of the sensor, and thus it *can be combined* with other methods aimed to directly enhance the sensor's speed, for example by employing nanoparticles with reported faster kinetics than Pd (e.g. PdAu, PdCo and PdTa alloys) or by utilizing polymer coatings with higher kinetics-enhancements such as PTFE. We have touched upon this outlook in our Conclusion and now we added also the additional alternative of using PTFE.

"The genericity of our strategy allows it to be combined with other optimization approaches, including the use of more sensitive transduction materials such as PdAu,^{2,19,52,53} (eightfold more sensitive than Pd at low H₂ concentrations) or PdTa⁵⁴ alloys, advanced data fittings capable of producing lower signal noise,⁵⁵ and with sensor designs aimed at increasing detection speed such as the use of nanoparticles with faster H₂ sorption kinetics (e.g. PdAu,^{2,19,52,53} PdCo,²⁴ PdTa⁵⁴) and of coating layers with higher kinetic-enhancement effects (e.g. PTFE,² twice as high as PMMA)."

And last, regarding the reproducibility, we now added data where we exposed our sensor to three cycles of 250 ppb H₂. It is clear that our sensor's response is reproducible even at this lowest concentration, with variance in absolute response in the order of the signal noise (*i.e.* ~0.01 nm).

Supplementary Figure 14. (a) $\Delta\lambda_{peak}$ response to three consecutive cycles of 250 ppb H₂ (grey areas). A reversible and reproducible sensor response to such low concentration of H₂ is observed. (b) Average sensor signal to the three cycles of 250 ppb H₂ exposure. An uncertainty of ~0.01 nm is recorded, which is in the same order of the sensor's signal noise.

We now refer to this Figure in the main text as:

"Due to this small noise, the sensor is able to measure even the lowest 250 ppb pulse (Supplementary Figure 14),..."

Reviewer 2

In this Communication, authors Nugroho et al. use an iterative algorithmic process to optimize an array-based hydrogen sensor. Tweaking parameters such as nanodisk diameter, nanodisk height, nanodisk pitch distance, and filtering layer (PMMA) thickness, the authors sought to optimize a figure of merit, and, by extension, minimize the hydrogen limit of detection (LoD). While the method of analysis is novel, and may be a significant contribution, this can not be concluded from the data presented here, as detailed below.

This work is not at the level of quality normally associated with Nature Commun. It may be publishable in a more specialized journal after major revision, including the addition of key data. The major issues are the following:

We highly appreciate the Reviewer's positive assessment of our method to increase the sensitivity in plasmonic gas sensors and of the potential contribution of our work. We are convinced that our work, together with the extensive additional data provided with this review, fully qualifies for the stringent publication standard of Nature Communications. Our approach and findings are not only important for the (hydrogen) sensing community but will also be of interest to the broader fields of fundamental and applied plasmonics. Below we address all Reviewer's concerns regarding our claims.

1. While low LoDs are potentially important in terms of understanding mechanistic aspects of sensor function, in any practical application required for hydrogen sensing, detection of the gas in a reactive atmosphere - such as air or humid air (in the case of safety sensors) - is absolutely required. In this paper, sensing was accomplished entirely with dry argon as the backing gas. Prior work has already demonstrated that air is a far more challenging medium for H₂ sensing than inert gases such as N₂ and Ar - especially as it concerns sensitivity and LoD. Papers, such as this one, that choose an expedient medium for sensor testing are routinely rejected from ACS Sensors, based upon the requirement that *meaningful media* for all types of sensors are an absolute requirement. This is simply not the case here. The authors write, "We expect such sensitivity to also hold in air, thanks to the excellent O₂ sieving provided by PMMA". These data should be added in revision.

We agree with the Referee that sensing hydrogen in air background is challenging, in particular when using Pd as the transduction material, because the O₂ in air will instead consume the H₂ on the surface of Pd to form water (H₂O). To circumvent such problem, one of the emerging solutions in the field is to employ one or more sieving overlayers, that block O₂ either *via* steric hindrance or through their low permeability toward O₂. Examples of these sieving materials include metal-organic frameworks (MOFs, for example <https://pubs.acs.org/doi/abs/10.1021/acsnano.7b04529>) and polymers (for example <https://www.nature.com/articles/s41563-019-0325-4>).

Here, we demonstrate that our inverse-designed optimized sensor can also benefit from the tandem polymer concept to retain its sensitivity in air, as we have shown in <https://www.nature.com/articles/s41563-019-0325-4>. We achieve this air resistance using poly(vinyl alcohol), PVOH, known for its low O₂ permeability. PVOH and PMMA have sufficiently similar optical properties to not interfere with the optical extinction spectra of our sensors. Using a 5 nm thick PVOH layer on top of the PMMA (and maintaining the optimized total polymer thickness of 300 nm) leads to a tandem sensor with essentially identical optical properties and FoM to the ones of the optimized sensor (*i.e.* with only PMMA), as can be seen by a comparison between the figure below and Figure 3d in the main text. The figure below is now included in the SI.

Supplementary Figure 17. (a) Experimental refractive indices of PMMA and PVOH. The data of PMMA is reproduced from²³. (b) FDTD-calculated extinction spectra of tandem sensor (see the schematic in Fig. 5b) for Pd (light gray) and PdH_{0.12} (dark grey) nanodisk arrays. The spectra are basically identical to the ones of the sensor coated with 300 nm PMMA (cf. Fig. 3d).

As shown in the Figure below, when we expose the tandem sensor to H₂ mixed with synthetic air, we indeed achieve a similar sensitivity as for the optimized sensor in Ar, and thus demonstrate the possibility to use our method for sensors operating in realistic environments. This data also provides an important insight on how one can still add targeted functionalities through addition of other polymer layer(s) without compromising the FoM/sensitivity optimized by the PSO algorithm.

Figure 5. Detection in air using a tandem polymer concept. (a) $\Delta\lambda_{peak}$ response to stepwise decreasing H₂ concentration (1000 to 0.25 ppm) with synthetic air at room temperature. Inset: zoomed-in version of the sensor response to 250 ppb H₂. (b) Measured $\Delta\lambda_{peak}$ as a function of H₂ concentration derived from (a). Gray dashed line is a guide to the eye. Light-gray symbols are the response of the optimized sensor (*i.e.* without PVOH) in Ar (cf. Fig 4b). The red dashed line marks the 3 σ value (0.03 nm).

Inset: Schematic of the tandem sensor comprising of a 5 nm PVOH film, acting as an O₂ barrier, on 295 nm thick PMMA layer. The PVOH-coated sensor shows a similar response for 250 ppb of hydrogen in air as the response of the optimized sensor under 250 ppb of hydrogen in Ar.

We now include the Figure to the main text and add the following discussion:

“As a final step to demonstrate the applicability of our strategy in realistic gas environments, that is, to detect H₂ in air, we apply the concept of tandem polymers with different functionalities^{2,53}. Such concept allows us to utilize multiple (polymer) layers that independently provide targeted functionalities, such as to efficiently block O₂ molecules. For this purpose, we use poly(vinyl alcohol), PVOH, known for its very low O₂ permeability and thus widely used as efficient O₂ barrier. To maintain the optimized polymer thickness of 300 nm in our sensor, we etched the existing PMMA film by 5 nm (which was crucially required to deposit PVOH, see Methods) and subsequently compensated such loss with a 5 nm thick PVOH layer (Fig. 5). Thanks to the similar refractive indices of PMMA and PVOH and the small PVOH layer thickness, the extinction spectra and the corresponding FoM of the tandem sensor are practically identical to the ones of the optimized sensor coated only by PMMA (Supplementary Figure 17). Consequently, the tandem sensor exposed to decreasing H₂ concentrations (1000 ppm to 250 ppb) in synthetic air exhibits a very similar response to the optimized sensor in Ar (Fig. 5).”

2. Nugroho, F. A. A. *et al.* Metal–polymer hybrid nanomaterials for plasmonic ultrafast hydrogen detection. *Nature Materials* **18**, 489–495 (2019).

53. Xie, B. *et al.* Metal Nanocluster—Metal Organic Framework—Polymer Hybrid Nanomaterials for Improved Hydrogen Detection. *Small* 2200634 (2022) doi:10.1002/SMLL.202200634.

We also added the experimental details regarding the PVOH deposition in the Methods:

“To produce the tandem sample, the sensor with PMMA was first etched in oxygen plasma for 2 s (50 W RF-power, 250 mTorr chamber pressure, and 40 sccm gas flow in a BatchTop Reactive Ion Etcher, PlasmTherm), to introduce hydrophilicity to the surface so that the PVOH solution can be dropcasted on it, which also resulted in the reduction of the PMMA thickness by ~5 nm according to the etch rate determined before.⁶³ Following that, a PVOH solution (0.1 wt.% in water) was spincoated (5000 rpm, 60 s) and then baked at 80 °C for 5 min. The obtained thicknesses were confirmed by ellipsometry (J.A. Woollam M2000).”

63. Nugroho, F. A. A., Albinsson, D., Antosiewicz, T. J. & Langhammer, C. Plasmonic Metasurface for Spatially Resolved Optical Sensing in Three Dimensions. *ACS Nano* **14**, 2345–2353 (2020).

and also the related FDTD simulation:

“For the tandem sample, the permittivity of PVOH was obtained from an ellipsometry measurement (J.A. Woollam M2000, Supplementary Figure 17).”

2. Likewise, extrapolation from higher H₂ partial pressures to very low LoD values, which are then quoted as an LoD, is unacceptable. If the point of the paper is ppb sensitivity, then please demonstrate measurements in this concentration range with some degree of robustness (that is to say, multiple exposures of the sensor to these concentrations so that the reproducibility of the response can be assessed). A 1 ppb LoD is mentioned at some point, and this is miles away from the value supported by the data provided in the ms.

Reviewer #1 had a similar concern on reproducibility of our sensor in resolving low concentration. To ensure that this measurement is robust, we have now added data showing our sensor’s response to three consecutive exposure to 250 ppb, the lowest detectable concentration of our sensor. As shown in Figure below, our sensor is robust even at this concentration range and this further strengthens our claim of 250 ppb detection.

Supplementary Figure 14. (a) $\Delta\lambda_{peak}$ response to three consecutive cycles of 250 ppb H₂ (grey areas). A reversible and reproducible sensor response to such low concentration of H₂ is observed. (b) Average sensor signal to the three cycles of 250 ppb H₂ exposure. An uncertainty of ~ 0.01 nm is recorded, which is in the same order of the sensor’s signal noise.

We now referred this Figure in the main text as:

“Due to this small noise, the sensor is able to measure even the lowest 250 ppb pulse (Supplementary Figure 14),...”

Regarding the detection limit, we are puzzled by the Reviewer’s remark as we never claim “a 1 ppb LoD” in our manuscript. Instead, we *demonstrate* 250 ppb detection (Figure 4a, and added Figure above), and mention the projected LoD of 200 ppb, corresponding to the analyte concentration at three times the signal noise. We explain this procedure in the text when we write:

“...and thus higher limits of detection (LoD), defined as the lowest analyte concentration measurable with a signal larger than 3σ .”

This definition of LoD is the standard procedure in the sensing literature and used by a number of international bodies (for example European and US Pharmacopeia). Even with this extrapolated LoD, we have been very careful with our claim, and therefore only highlight the demonstrated *ppb* detection when we mention our sensitivity in the manuscript, as written in the original Abstract

“Guided by a particle swarm optimization algorithm, we numerically identify and experimentally demonstrate a sensor with an optimal balance between a narrow spectral linewidth and a large field enhancement inside the nanoparticles, enabling a measured hydrogen detection limit of 250 parts-per-billion (ppb).”

and in the end of the Introduction

“This generic approach, which can benefit any direct plasmonic sensing platform, guides us to identify and experimentally demonstrate a sensor nanoarchitecture with a discernible signal down to 250 ppb; the lowest detection limit reported for an optical hydrogen sensor.”

3. A important issue with the presentation of the sensing data is that the exposures to H₂ are carried out *in order of decreasing concentration*. This practice obscures any "memory effect" of the sensing elements. It is well understood that proper practice is to carry out these exposures in random order of H₂ concentration, with the exposure to several concentrations across the calibration curve repeated more than once so that the reproducibility of the sensor response can be assessed - after exposures to a range of different H₂ concentrations. This practice of presenting a train of monotonically decreasing (or increasing) H₂ exposures (increasing common!) should be rejected by journals of the stature of the Nature journals.

We forgot to mention that it has been our practice to always expose our sensors, prior to any measurement, with multiple cycles (>20 times) of high pressure of hydrogen (1 bar) followed by desorption in vacuum. This procedure, which has been recently detailed by Alekseeva *et al.* (<https://www.nature.com/articles/s41467-021-25660-x>), avoids residual memory effects due to the hydrogen-induced restructuring of Pd nanostructures, and therefore the corresponding signal change, upon H₂ ab/desorption. We add now this additional experimental detail in the Method section.

“Prior to measurements, all sensors were exposed to multiple cycles (>20) of pure H₂ (1 bar) and vacuum at room temperature to stabilize the hydrogen-induced microstructural changes in the nanoparticles.⁶⁶”

66. Alekseeva, S. et al. Grain-growth mediated hydrogen sorption kinetics and compensation effect in single Pd nanoparticles. *Nature Communications* 12, 5427 (2021).

We agree with the Reviewer that in the absence of a sensor pretreatment, a monotonically varying pressure may carry the signature of memory effects. However, with our pretreatment this issue is no longer a problem, as now we confirm with the additional data requested where we expose the sensor to a series of random H₂ concentration.

“Supplementary Figure 13. (a) $\Delta\lambda_{peak}$ response to stepwise random H₂ concentration (250 to 0.25 ppm) in Ar carrier gas at room temperature. Inset: zoomed-in version of the sensor response to 250 ppb H₂. (b) Measured $\Delta\lambda_{peak}$ as a function of H₂ concentration derived from (a). The transparent symbols and gray dashed line are reproduced from Fig. 4b. The sensor’s responses to these random H₂ exposure are consistent with the descending one, and thus exemplifying the reproducibility of the sensor.”

With this data in hand, however, we opt to keep the descending-pressure data in the main text as we believe that monotonically varying pressures to be a systematic and pedagogic way to demonstrate a response from a (gas) sensor, especially when ones aim to determine the LoD, and they have been shown widely in the literature, including very recently in Nature journals (<https://www.nature.com/articles/s41563-019-0325-4> and <https://www.nature.com/articles/s41467-021-22697-w>).

4. Baseline drift is a major issue in the data presented in Fig. 4. Especially in dry Ar gas, this is surprising. What is the origin of this drift? Is it caused by H₂ exposures, or is the baseline drifting before H₂ exposures are initiated? Presumably, these data are amongst the best acquired by the authors, meaning that data sets NOT showing baseline drift were not available. The practical utility of any gas sensor exhibiting this characteristic is very limited.

We believe the reviewer is referring to a drift of ~0.25 nm in the baseline of our measurements occurring midway through an experiment running for 26 hours. Such a peak shift corresponds to 0.08% variation with respect to the peak linewidth of ~296 nm (see Figure 3d in the manuscript). This is not at all unreasonable and likely due to inevitable small adjustments or vibrations in the optical setup (see for example Figure 5 in our previous work <https://doi.org/10.1039/C8NR03751E>).

We should also mention that we always wait for a stable baseline before each hydrogen exposure. Hence, it is unambiguous that any increase in the sensor response is due to hydrogen exposure. In fact, all of the sensor's responses to hydrogen in the time period when the small baseline drift occurs are significantly larger than the drift.

Looking also at other data, for example for control sensor in Figure 4c, the baseline is pretty much stable throughout the experiment. And now with also additional data we provide as the response to Question #2, it is clear that achieving stable baseline is not impossible and that the conclusions we draw from our work are not affected in any way by the small baseline drift.

We now add this sentence in the caption of Figure 4 to discuss about the origin of the drift:

“The slight baseline drift likely arises from minor adjustments in the setup over the course of the experiment.”

Reviewer 3

1. In this paper the authors used an inverse nanophotonic design approach to identify and experimentally demonstrate an ultrasensitive plasmonic hydrogen detector based on collective resonances in periodic arrays of palladium nanoparticles. The measured ppb limit of detection seems to be an order of magnitude lower than previously optical hydrogen sensor.

the idea to optimize the array configuration in order to achieve very high sensitivity is not new and has been reported recently in ACS Sens. 2020, 5, 4, 917–927 (not cited in this paper). The original approach used by the authors here comprised a population-based stochastic evolutionary computation technique. This sounds actually interesting because it can be potentially applied to many other applications.

We thank the Reviewer for the positive comment and for recognizing the potential of our method for broad applications. Related to the invoked paper, although it is similar in the sense that it employed periodical Pd arrays, the sensing configuration used, however, is different. In the paper, the authors use the so-called perfect absorber configuration, and thus did not specifically aim to increase the sensitivity *via* linewidth reduction of a surface lattice resonance, as in our case. It is also worth mentioning that the referred paper only achieves 100 ppm detection limit, a target that could potentially benefit from our design approach *via* inverse nanophotonic optimization. As the Reviewer also observes, our optimization strategy is not unique to surface lattice resonances and can be applied to other sensing platforms, including the perfect absorbers used in the invoked paper, or nanoparticles on mirror. We have added this short discussion in the conclusions:

“Our inverse design approach, however, also permits the optimization of nanoparticle arrays for sensing platforms using different configurations such as perfect absorbers²⁰ and nanoparticles-on-mirror,⁵⁶ and different readouts such as single-wavelength mode devices,^{57,58} opening the door to low-cost, ultrasensitive platforms.”

20. Sterl, F. *et al.* Design Principles for Sensitivity Optimization in Plasmonic Hydrogen Sensors. *ACS Sensors* **5**, 917–927 (2020).

56. Tittl, A. *et al.* Plasmonic Smart Dust for Probing Local Chemical Reactions. *Nano Letters* **13**, 1816–1821 (2013).

2. The other noteworthy aspect/result in the paper is the obtained detection limit (sub ppm). Anyway, although this is an order of magnitude better with respect to other examples in literature, the way it is calculated is not convincing. The authors should report (or eventually perform) H₂ exposure also starting from the lower concentrations (i.e. from 0 up to 1000ppm). the delta lambda of 0.05nm (0.5 Ang!!) sounds very hard to be reproducible. moreover, it is known that H₂ exposure can change the volume of Pd nanostructures and this can impact in the resonance of the system (for this reason is important to test the system starting from 0 and increasing the concentration) (this effect is for example discussed in *Optics Express* **28**, , pp. 25383-25391 (2020); and *Journal of Alloys and Compounds* **704**, 2017, 303-310)

We first want to discuss the possibility of resolving 0.05 nm in our measurement. To this end, achieving 0.01 nm noise in a plasmonic sensing setup has become a routine in the field. Hence, in theory, resolving a shift of 0.05 nm (that is, 5 times the signal noise) is possible. Such feat is particularly easy when using Au/Ag nanostructures with its inherent narrow linewidth. Here we want to restate our novelty in that we can achieve such low noise in Pd nanostructures by narrowing its linewidth *via* surface lattice resonances while maintaining high response towards hydrogen (in surface lattice resonances the narrower the linewidth the less field is contained in the nanostructures, hence the lower sensitivity for a change inside them, as we discussed in the manuscript), efficiently *via* an optimization algorithm. With this approach, we now make it possible for Pd nanostructures to resolve 0.05 nm. With now additional data below, where we expose our sensor to three consecutive 250 ppb H₂, it is clear that our method is reproducible.

Supplementary Figure 14. (a) $\Delta\lambda_{peak}$ response to three consecutive cycles of 250 ppb H_2 (grey areas). A reversible and reproducible sensor response to such low concentration of H_2 is observed. (b) Average sensor signal to the three cycles of 250 ppb H_2 exposure. An uncertainty of ~ 0.01 nm is recorded, which is in the same order of the sensor's signal noise.

Related to the change of the Pd nanostructures upon exposure to H_2 , we first want to repeat the answer given to Reviewer #2 for his/her similar concern. Here we forgot to mention that it has been our practice to always expose our sensors, prior to any measurement, with multiple cycles (>20 times) of high pressure of hydrogen (1 bar) followed by desorption in vacuum. This procedure, which has been recently detailed by Alekseeva *et al.* (<https://www.nature.com/articles/s41467-021-25660-x>), avoids residual memory effects due to the hydrogen-induced restructuring of Pd nanostructures, and therefore the corresponding signal change, upon H_2 ab/desorption. We add now this additional experimental detail in the Method section.

“Prior to measurements, all sensors were exposed to multiple cycles (>20) of pure H_2 (1 bar) and vacuum at room temperature to stabilize the hydrogen-induced microstructural changes in the nanoparticles.”⁶⁶

66. Alekseeva, S. *et al.* Grain-growth mediated hydrogen sorption kinetics and compensation effect in single Pd nanoparticles. *Nature Communications* 12, 5427 (2021).

With this pretreatment, there will be no effect in the sensors when we measured them in decreasing, increasing, or even random H_2 concentration as the nanoparticles are already stable and thus absorb/desorb H_2 reversibly, as we now demonstrate in the Figure below.

“Supplementary Figure 13. (a) $\Delta\lambda_{peak}$ response to stepwise random H_2 concentration (250 to 0.25 ppm) in Ar carrier gas at room temperature. Inset: zoomed-in version of the sensor response to 250 ppb H_2 . (b) Measured $\Delta\lambda_{peak}$ as a function of H_2 concentration derived from (a). The transparent symbols and gray dashed line are reproduced from Fig. 4b. The sensor's responses to these random H_2 exposure are consistent with the descending one, and thus exemplifying the reproducibility of the sensor.”

Additionally, with respect to the change in the volume of Pd, in the H₂ range we studied, the corresponding volume change to H₂ sorption is negligible, as it is *far* below the hydride β -phase formation where the volume expands significantly (in Pd at room temperature this occurs around 2-3% H₂ – our highest investigated concentration is 0.1% H₂). Indeed we agree that exposure at high H₂ concentration may cause Pd hydrogen sensors to deteriorate due to cracking and/or peeling, however this is particularly true for *thin film* sensors, as shown in the papers mentioned by the Reviewer. Hence, thanks to our pretreatment, our H₂ range well below the α -to- β phase transition, and our use of nanoparticles, we are certain that the expansion-related issue is negligible in our sensors. In fact, when we run our FDTD simulation for the PSO algorithm, we deliberately chose to simulate Pd hydride of 0.125 to avoid any error related to the volume change in the nanoparticles, since the Pd is still in the α -phase and the volume expansion is negligible. We wrote this rationale in the Methods in our original manuscript:

“Here PSO was employed to optimize the structural parameters of the plasmonic hydrogen sensor to yield the highest FoM defined by Eq. 1. To this end, we chose to use PdH_{0.125} for the calculation of the hydride phase for the following reasons: (i) This is the lowest Pd hydride concentration whose dielectric function is available in the literature.²⁵ (ii) At this concentration, the Pd hydride is still at the diluted α -phase, with negligible lattice expansion. This condition prevents inaccurate calculation during FDTD simulation where the expansion of the nanodisk has to be included.⁶⁵ (iii) The chosen hydride concentration is also in line with the targeted range of the hydrogen concentration. (iv) Lastly, the accompanied spectral change of the sensor at this hydride concentration was expected to be small enough so that it would be the same SLR peak that was considered, thus avoiding false $\Delta\lambda_{peak}$ determination when calculating the FoM, as we detailed later below.”

3. Finally, the authors should discuss how this value can be measured in a real on-field application.

We acknowledge that real application using optical setup we use here is not very practical as it involves expensive broadband spectrophotometer. However, as we argue in our conclusions, our optimization routine can easily be extended to other figures of merit specific to single-wavelength detectors.

“Our inverse design approach, however, also permits the optimization of nanoparticle arrays for sensing platforms using different configurations such as perfect absorbers²⁰ and nanoparticles-on-mirror,⁵⁶ and different readouts such as single-wavelength mode devices,^{57,58} opening the door to low-cost, practical,⁵⁹ ultrasensitive platforms.”

59. Herkert, E., Sterl, F., Strohfeldt, N., Walter, R. & Giessen, H. Low-Cost Hydrogen Sensor in the ppm Range with Purely Optical Readout. ACS Sensors 5, 978–983 (2020).

In fact, plasmonic optical gas sensors are currently used in real on-field applications (Goteborg, NO₂ detection). These are single wavelength (as in <https://pubs.acs.org/doi/abs/10.1021/acssensors.9b02314>) which are simpler and cheaper to make, only requiring an LED and a photodiode.

4. With respect to the state-of-the-art, the authors missed to mention several important papers in the references (among the others - actually there are tons of papers on H₂ sensing by means of Pd nanostructures):

ACS Sens. 2020, 5, 4, 917–927

Nano Energy 71, 22020, 104558

Nanophotonics, vol. 3, no. 3, 2014, pp. 157-180

MRS Bulletin volume 38, 495–503 (2013)

Nature Materials volume 10, 631–636 (2011)

Sensors and Actuators B: Chemical 295, 15, 2019, 101-109

Nanomaterials 2021, 11, 3100

We thank Referee for the suggestions. Indeed, there are numerous reports on the use of Pd nanostructures as H₂ sensors, especially in the past years; a fact that makes us believe that our results here will be timely and important. Having said that, we certainly cannot include everything due to the limitation in the reference number and we have tried our best to include those that are relevant to the discussions we have. Nonetheless, we agree with the Reviewer that there are important and relevant works to be cited, and hence we include some of the suggested references. In particular:

“Emerging examples of direct plasmonic sensors are plasmonic hydrogen sensors based on palladium (Pd) nanoparticles and their alloys.^{2,18–21}”

“Our inverse design approach, however, also permits the optimization of nanoparticle arrays for sensing platforms using different configurations such as perfect absorbers²⁰ and nanoparticles-on-mirror,⁵⁶ and different readouts such as single-wavelength mode devices,^{57,58} opening the door to low-cost, practical,⁵⁹ ultrasensitive platforms. Beyond hydrogen sensing, our approach can be extended to arrays of surface-functionalized nanoparticles with resonances that are sensitive to the adsorption of specific gasses via refractive index effects or chemical interface damping,^{60,61} with the potential to address a wider range of societal needs, from home safety to urban air pollution monitoring.⁶²”

20. Sterl, F. *et al.* Design Principles for Sensitivity Optimization in Plasmonic Hydrogen Sensors. *ACS Sensors* **5**, 917–927 (2020).

21. Luong, H. M. *et al.* Bilayer plasmonic nano-lattices for tunable hydrogen sensing platform. *Nano Energy* **71**, 104558 (2020).

56. Tittl, A. *et al.* Plasmonic Smart Dust for Probing Local Chemical Reactions. *Nano Letters* **13**, 1816–1821 (2013).

61. Tittl, A., Giessen, H. & Liu, N. Plasmonic gas and chemical sensing. *Nanophotonics* **3**, 157–180 (2014).

REVIEWERS' COMMENTS

Reviewer #1 (Remarks to the Author):

I still see the slow response times as a major drawback/limitation for this sensor especially for publication in a high impact journal like Nature Communication. Nevertheless, the authors have satisfactorily addressed all my concerns on the previous version of the manuscript, including conducting additional experiments related to response times and selectivity of the sensor in the presence of other gases (except for the effects of moisture).

Reviewer #3 (Remarks to the Author):

the revisited version of the manuscript is significantly improved and the authors replied in a very convincing way to all the referees' comments.

I don't see now any particular criticism.

I recommend the publication in the present form.

We are very happy for the positive responses of the reviewers to our revision. Below, we have addressed the one remaining concern.

Point-to-Point Response

Reviewer 1

I still see the slow response times as a major drawback/limitation for this sensor especially for publication in a high impact journal like Nature Communication. Nevertheless, the authors have satisfactorily addressed all my concerns on the previous version of the manuscript, including conducting additional experiments related to response times and selectivity of the sensor in the presence of other gases (except for the effects of moisture).

We thank the Reviewer for his/her very positive assessment and inputs during the revision. Regarding the slow response time, in the last revision we have proposed how one can improve such aspect in parallel with the achieved high sensitivity in our sensors. In particular we wrote in the Supplementary Information

“To deduce the response and recovery times of the sensors we use the commonly used t_{90} and t_{10} , respectively (Supplementary Fig. 15a-b). As plotted in Supplementary Fig. 15c-d, the response and recovery times of the sensor increase with the lowering H_2 concentration. Such observation is inherent to Pd nanostructures as previously shown.^{15,17,20} As a result, at the lowest H_2 concentration of 250 ppb, the sensor’s response time is in the order of 40 min. Interestingly, the response and recovery times of the control random array sensor are practically similar to the optimized one (Supplementary Fig. 15c-d). Such finding reveals that our method of increasing the sensor’s sensitivity via periodic arrangement does not affect its sensing speed as it is mainly defined by the materials design.”

and in Discussion

“The genericity of our strategy allows it to be combined with other optimization approaches, including the use of more sensitive transduction materials such as PdAu,^{2,19,25,54} (eightfold more sensitive than Pd at low H_2 concentrations) or PdTa⁵⁵ alloys, advanced data fittings capable of producing lower signal noise,⁵⁶ and with sensor designs aimed at increasing detection speed such as the use of nanoparticles with faster H_2 sorption kinetics (e.g. PdAu,^{2,19,25,54} PdCo,²⁴ PdTa⁵⁵) and of coating layers with higher kinetic-enhancement effects (e.g. PTFE,² twice as high as PMMA).”

Nonetheless, to address this current limitation explicitly, we have added a brief discussion in the manuscript:

“Regarding response/recovery times, the use of a thick PMMA film in our sensor significantly slows down the kinetics, with a response time of ~40 min at 250 ppb.

However, as we discuss in the Supplementary Information and also later below, such speed can be increased by incorporating materials with faster hydrogenation kinetics and/or optimized nanostructure geometries.”

Reviewer 3

the revisited version of the manuscript is significantly improved and the authors replied in a very convincing way to all the referees' comments. I don't see now any particular criticism. I recommend the publication in the present form.

We thank the Reviewer for the recommendation to publish, and for the inputs she/he gave that definitely improved the manuscript.